# How Reliable Are AI-Generated-Text Detectors? An Assessment Framework Using Evasive Soft Prompts

**Tharindu Kumarage**    **Paras Sheth**    **Raha Moraffah**    **Joshua Garland**    **Huan Liu**
Arizona State University
{kskumara, psheth5, rmoraffa, jtgarlan, huanliu}@asu.edu

## Abstract

In recent years, there has been a rapid proliferation of AI-generated text, primarily driven by the release of powerful pre-trained language models (PLMs). To address the issue of misuse associated with AI-generated text, various high-performing detectors have been developed, including the OpenAI detector and the Stanford DetectGPT. In our study, we ask how reliable these detectors are. We answer the question by designing a novel approach that can prompt any PLM to generate text that evades these high-performing detectors. The proposed approach suggests a universal evasive prompt, a novel type of soft prompt, which guides PLMs in producing "human-like" text that can mislead the detectors. The novel universal evasive prompt is achieved in two steps: First, we create an *evasive soft prompt* tailored to a specific PLM through prompt tuning; and then, we leverage the transferability of soft prompts to transfer the learned evasive soft prompt from one PLM to another. Employing multiple PLMs in various writing tasks, we conduct extensive experiments to evaluate the efficacy of the evasive soft prompts in their evasion of state-of-the-art detectors.

## 1 Introduction

Recent advances in transformer-based Pre-trained Language Models (PLMs), such as PaLM (Thoppilan et al., 2022), GPT 3.5 (Ouyang et al., 2022), and GPT 4 (OpenAI, 2023), have greatly improved Natural Language Generation (NLG) capabilities. As a result, there is a proliferation of highly compelling AI-generated text across various writing tasks, including summarization, essay writing, academic and scientific writing, and journalism. While AI-generated text can be impressive, it also brings potential risks of misuse, such as academic fraud and the dissemination of AI-generated misinformation (Dergaa et al., 2023; Kreps et al., 2022). Consequently, to combat the misuse associated with

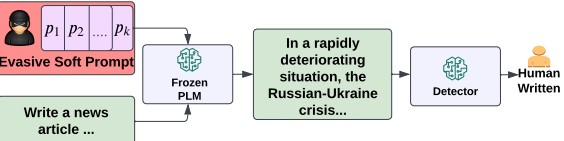

Figure 1: Evasive soft prompt guiding PLM to evade AI-generated-text detector

AI-generated text, we observe the emergence of automatic AI-generated-text detectors, which include commercial products like GPTZero (Tian, 2023), Turnitin AI text detector [1], as well as open-source methods as DetectGPT (Mitchell et al., 2023) and fine-tuned versions of OpenAI GPT-2 detector (Solaiman et al., 2019).

Although AI-generated-text detectors exhibit impressive performance during standard evaluations using existing datasets, their efficacy can be challenged when deployed in real-world scenarios. For instance, malicious users may manipulate the PLMs to generate text in a way that misleads the detector into classifying it as human-written. Such scenarios can lead to detrimental consequences in practical applications like academic fraud detection or AI-generated news identification. Consequently, it is imperative to assess the reliability of current AI-generated-text detectors in such situations. Therefore, in this study, we aim to answer the question of how reliable the existing state-of-the-art AI-generated-text detectors are.

To answer this question, we propose an evasive soft prompt, a novel form of soft prompt specifically crafted to enable the evasion of AI-generated-text detectors. As illustrated in figure 1, when combined with a standard input prompt, this evasive soft prompt effectively guides the PLM towards generating texts that convincingly resemble human-written content, thus evading detection by AI-generated-text detectors. Given the rapid ad-

---

[1]https://www.turnitin.com/solutions/ai-writing

vancements of PLMs in recent times, the development of a more generalized framework that can assess the reliability of the detectors on both current and future PLMs is very crucial. To facilitate this, we design our evasive soft prompt scheme to be "universal", making it feasible for any PLM to adopt it easily. The proposed universal evasive soft prompt is achieved through a two-step process. First, we develop an evasive soft prompt tailored for a particular PLM using prompt tuning (PT) (Lester et al., 2021a). Then, leveraging the transferability of soft prompts, we efficiently transfer the acquired evasive soft prompt from one PLM to another. We refer to our framework for **E**vasive **S**oft **P**rompts for AI-generated-text detectors as **EScaPe**.

Our experiments on different PLMs and diverse writing tasks provides empirical evidence of the efficacy of **EScaPe** in evading state-of-the-art detectors. Furthermore, our experiments on transferability demonstrate the successful application of evasive soft prompts across multiple prominent PLMs. Additionally, we present the effectiveness of **EScaPe** in transferring across different state-of-the-art detectors. Our findings underscore the importance of further research in designing more reliable detection mechanisms for combating the misuse related to AI-generated text. In summary, our study makes the following key contributions:

1. We propose the framework **EScaPe**, which introduces the concept of evasive soft prompts—a novel type of soft prompt specifically designed to guide PLMs in evading state-of-the-art AI-generated-text detectors.

2. We demonstrate the transferability of evasive soft prompts learned through the **EScaPe** framework across different PLMs, making it a universal evasive soft prompt to assess the reliability of AI-generated-text detectors on any current or future PLM.

3. We conducted extensive experiments by employing a wide range of PLMs in various real-world writing tasks to demonstrate the effectiveness of the proposed framework, consequently highlighting the vulnerability of the existing AI-generated-text detectors.

## 2 Related Work

### 2.1 Detection of AI-Generated Text

The task of AI text detection is generally considered a binary classification task, with two classes:

"Human-written" and "AI-written." In the early days, several supervised learning-based classification methods were explored for detecting AI-generated text, such as logistic regression and SVC (Ippolito et al., 2019). In contrast, GLTR (Gehrmann et al., 2019) employs a set of simple statistical tests to determine whether an input text sequence is AI-generated or not, making the method zero-shot. In recent years, fine-tuned PLM-based detectors have emerged as the state-of-the-art, including OpenAI's GPT2 detector (Solaiman et al., 2019; Jawahar et al., 2020; Zellers et al., 2019; Kumarage et al., 2023). With the rapid advancement of newer large language models, there is an increasing emphasis on the capabilities of few-shot or zero-shot detection and the interpretability of these detectors (Mitrović et al., 2023). Some new detectors include commercial products such as GPTZero (Tian, 2023), and OpenAI's detector (Kirchner et al., 2023). A recent high-performing zero-shot detection approach called DetectGPT (Mitchell et al., 2023) operates on the hypothesis that minor rewrites of AI-generated text would exhibit lower token log probabilities than the original sample. Watermarking on PLM-generated text is also an exciting approach gaining attention in the research community (Kirchenbauer et al., 2023). However, it assumes that the AI generator itself supports the implementation of watermarking, which reduces the practicality of the approach.

### 2.2 AI-generated-text Detector Evasion

Few recent studies have investigated the efficacy of paraphrasing as a technique for evading AI-generated text detection. (Sadasivan et al., 2023; Krishna et al., 2023) demonstrated that paraphrasing the text can considerably undermine the performance of AI-generated text detectors, raising concerns about such detection methods' reliability. However, our work differs significantly from paraphrasing for the following reasons: 1) we aim to assess the reliability of existing detectors against the capabilities of the original PLM that generated the text. In paraphrasing, a secondary PLM is used to rephrase the original PLM's text to evade detection, resulting in a two-step process, and 2) Paraphrasing attack evaluation provides a unified score for type I and type II errors of the detector. However, it is essential to distinguish between these two types of errors to validate the detector's reliability. In real-world scenarios, the most probable situation

would involve type II errors, where malicious actors attempt to generate AI text that can evade the detectors and result in a false negative. Our study focuses explicitly on emulating such a scenario and evaluating the type II errors.

## 2.3 Soft Prompt Tuning

Prompt tuning is a widely used approach for guiding PLMs to generate desired outputs (Jiang et al., 2020). GPT-3 demonstrated early success in prompt tuning, achieving remarkable performance on multiple tasks using tailored prompts (Brown et al., 2020). This led to extensive research on hard prompts, which are manually or automatically crafted prompts in discrete space (Mishra et al., 2021; Gao et al., 2020). Simultaneously, researchers have explored the potential of soft prompts (Liu et al., 2023). Unlike hard prompts, soft prompts are in the continuous embedding space. Therefore, soft prompts can be directly trained with task-specific supervision (Wu and Shi, 2022; Gu et al., 2022). Notable methods for soft prompts, such as prompt tuning (PT) (Lester et al., 2021b), and P-tuning (Liu et al., 2022), have achieved performance comparable to full parameter finetuning of PLMs for downstream tasks. Furthermore, recent works have presented the transferability of soft prompts across tasks and across PLM architectures (Su et al., 2022; Vu et al., 2021).

## 3 Methodology

Figure 2 illustrates the process of generating the universal evasive soft prompt, which involves two main steps: evasive soft prompt learning and evasive soft prompt transfer. In the first step, we learn an evasive soft prompt for a specific frozen PLM (source PLM -$PLM^s$). The second step is the evasive soft prompt transfer, which involves transferring the learned soft prompt to a frozen target PLM ($PLM^t$). In the subsequent sections, we comprehensively discuss these two steps.

### 3.1 Evasive Soft Prompt Learning

#### 3.1.1 Overview of Learning

Evasive soft prompt learning aims to tune the soft prompt $P^s$ so that once the learned $P^s$ is inputted into $PLM^s$, it generates text classified as "Human-written" by the detector. To accomplish this objective, our end-to-end learning framework is defined as follows: first, we configure the soft prompt $P^s$ based on the Prompt Tuning (PT) method (Lester

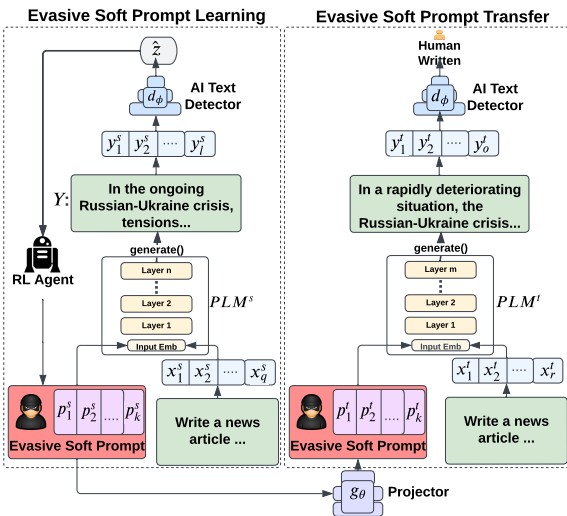

Figure 2: The proposed framework, **EScaPe**, encompasses learning evasive soft prompt on a specific PLM, followed by its transfer to a target PLM.

et al., 2021b). Second, we input both the soft and natural language prompt, which reflects the specific writing task (Figure 1 provides an example of a news writing task), into $PLM^s$, and generate the corresponding output text response. Next, we pass this generated text response to the AI-generated-text detector and obtain the probability values for the final classification classes. Finally, using the detector's output as the reward, we propagate the learning error and update the soft prompt $P^s$ using the Proximal Policy Optimization (PPO) method (Schulman et al., 2017; Ziegler et al., 2019).

#### 3.1.2 Configuring the Evasive Soft Prompt

We use the PT method for defining and configuring the evasive soft prompt on a given $PLM^s$. Here the parameters of $PLM^s$ are frozen, and the only parameter we intend to update is the soft prompt $P^s$. The soft prompt $P^s$ consists of a collection of $k$ virtual tokens $p_1^s, p_2^s, ..., p_k^s$ connected to the input of $PLM^s$. Unlike regular input tokens, these $k$ virtual tokens are learnable embedding vectors that can be fine-tuned to guide $PLM^s$ in accomplishing a specific task or goal. In our case, these learnable tokens aim to steer the generation of text that evades AI text detectors. Formally, given a natural language prompt sequence with $q$ tokens $X = x_1^s, x_2^s, ..., x_q^s$, we input $X$ into the PLM and get the corresponding embedding representations while prepending $k$ randomly initialized soft prompts $p_1^s, p_2^s, ..., p_k^s$ before them. Here, $p_i^s \in \mathbb{R}^{e_s}$, and $e_s$ is the input embedding size of $PLM^s$.

### 3.1.3 Text Generation

After configuring the evasive soft prompt for $PLM^s$, we generate the corresponding output text $Y^s$. This generation is conditioned on the soft prompt tokens $P^s = p_1^s, p_2^s, ..., p_k^s$ and the input tokens $X^s = x_1^s, x_2^s, ..., x_q^s$, as shown below:

$$p(Y^s) = \prod_{i=q}^{N} p(x_i^s | P^s, x_1^s...x_q^s...x_{i-1}^s) \quad (1)$$

In equation 1, $N$ represents the size of the generated output, determined by the stopping criteria used during text generation with $PLM^s$. In our case, we utilize the default maximum sequence size criteria, which halts the generation after reaching the maximum defined sequence length.

### 3.1.4 Reward Generation from the Detector

Given the text output generated by $PLM^s$, our objective is to calculate a reward value that indicates the degree to which the generated text resembles human-written text for AI text detectors. For this purpose, we utilize a proxy AI-generated-text detector. Given the generated text $Y$ from $PLM^s$, we compute the probabilities of $Y$ being classified as "Human" and "AI". Formally, we define the input text to the detector as $Y = (y_1^s, y_2^s, ..., y_l^s)$, where $(y_1^s, y_2^s, ..., y_l^s)$ represents the tokens of input text $Y$ based on the tokenizer of the detector model. The detector has learned the function $d_\phi$, which takes the input tokens and predicts the class probabilities.

$$\{cls_p^0, cls_p^1\} = d_\phi(y_1^s, y_2^s, ..., y_l^s)$$
$$\hat{z} = cls_p^0, \quad (2)$$

where $cls_p^i$ represents the probability for class $i$, and $\hat{z}$ denotes the probability of the "Human" class ($i = 0$). The ultimate objective of our framework is to fine-tune the soft prompt $P^s$ to maximize the reward $\hat{z}$. In other words, we aim to guide $PLM^s$ in generating text that enhances the "Human" class probability as predicted by the detector.

### 3.1.5 Finetuning via Reinforcement Learning

After obtaining the reward from the detector, $\hat{z}$, we utilize reinforcement learning to update our prompt's parameters effectively. Specifically, RL is used to incorporate the detector's feedback, i.e., probability scores assigned to the labels "human-written" and "AI-generated") to guide the generation. The RL module abides by the following structure:

- **State** consists of two components: the evasive soft prompt and the detector. The agent acts as an optimizer, updating the evasive soft prompt based on the detector's output.

- **Reward** is the reward function that aids the RL agent in performing actions more aligned towards the goal – making the generated text looks human-written to the detector. Given the language model $PLM^s$, we initialize the policy $\pi$ as $PLM^s$ and then fine-tune only the evasive soft prompt embedding $P^s$ to perform the task well using RL. We fit a reward model $f$ using the loss:

$$loss(f) = \mathbb{E}_{(Y,\{\hat{z}\}_i)} \left[ \log \frac{e^{f(Y,\hat{z})}}{\sum_i e^{f(Y,\hat{z_i})}} \right] \quad (3)$$

where $Y$ is the text generated by $PLM^s$ for the prompt $X$, and $\hat{z}$ is the detector output. To keep $\pi$ from moving too far from $PLM^s$, we leverage a penalty with expectation $\beta\text{KL}(\pi, PLM^s)$, (where KL denotes the KL divergence) modifying the RL reward as,

$$F(y, \hat{z}) = f(y, \hat{z}) - \beta\text{KL}(\pi, PLM^s) \quad (4)$$
$$= f(y, \hat{z}) - \beta \log \frac{\pi(y \mid x)}{PLM^s(y \mid x)} \quad (5)$$

Here $x \in X$ and $y \in Y$ (single instance of an input prompt and generated text). Adding the KL term adds two benefits. First, it prevents the policy from moving too far from the range where $f$ is valid, and second, it enforces coherence and topicality in the generated text.

- **Action** of the RL agent is updating the $k$ tokens of the evasive soft prompt $P^s$ based on the policy objective formulated as,

$$J(\lambda) = \mathbb{E}_{\pi_\lambda}[\sum_{i=1}^{k} F_i], \quad (6)$$

where $\lambda$ is the language model's parameters related to the soft prompt embedding, and $F_i$ is the reward for input $y_i$ and reward $\hat{z}_i$. We use the gradient of this objective after each iteration to update the evasive soft prompt.

Optimizing the evasive soft prompt based on the defined reward signals, the model finally outputs text that is detected as human-written by the detector.

## 3.2 Evasive Soft Prompt Transfer

After learning the evasive soft prompt on the $PLM^s$, we transfer the trained evasive soft prompt to the semantic space of target $PLM^t$. Following the cross-architecture transferability work of soft prompts (Su et al., 2022), we train a projector to efficiently map the evasive soft prompt from the source PLM to the target PLM.

Formally, we define the soft prompts of the source $PLM^s$ as $P^s = p_1^s, ..., p_k^s$; $P^s \in \mathbb{R}^{e_s \times k}$. The projector $g_\theta$ aims to project $P^s$ to $P^t \in \mathbb{R}^{e_t \times k}$, representing the semantic space of the target $PLM^t$. Here, $e_s$ and $e_t$ denote the input embedding dimensions of the source and target PLMs, respectively. We parameterize the projector function $g_\theta$ using a two-layer feed-forward neural network:

$$P^t = g_\theta(P^s) = W_2(\sigma(P^s W_1 + b_1)) + b_2 \quad (7)$$

In Equation (7), $W_1 \in \mathbb{R}^{e_h \times e_s}$, $W_2 \in \mathbb{R}^{e_t \times e_h}$ are trainable matrices, $b_1 \in \mathbb{R}^{e_h}$, $b_2 \in \mathbb{R}^{e_t}$ are biases, and $\sigma$ is a non-linear activation function. Here $e_h$ denotes the hidden layer size of the projector network. Finally, we train the above cross-model projection parameters using our RL framework for evasive soft prompt learning. However, given that we are now initializing the $P^t$ from an already learned evasive soft prompt $P^s$, learning the evasive soft prompt for the target $PLM^t$ is done in fewer iterations.

## 4 Experiment Setup

Here we describes the experimental settings used to validate our framework, including the AI Generators (PLMs), writing tasks (datasets), detection setup, baselines, and the implementation details of **EScaPe** to support reproducibility.

## 4.1 AI-Text Generators (PLMs)

We evaluate our framework on numerous open-source PLMs readily available on HuggingFace. Specifically, we selected the current state-of-the-art PLMs such as **LLaMA**(7B; (Touvron et al., 2023)), **Falcon**(7B; (Almazrouei et al., 2023)), as well as-established powerful PLMs like **GPT-NeoX**(20B; (Black et al., 2022)) and **OPT**(2.7B; (Zhang et al., 2022)). In our experiments, we used the above PLMs in two ways. Firstly, for zero-shot language generation, we generated AI text for different writing tasks considered in our analysis. Secondly, we used them as the base PLMs for our framework's evasive soft prompt learning task. We em-

ployed similar language generation parameters in both cases, setting the *top-p* to 0.96 and the *temperature* to 0.9.

## 4.2 Writing Tasks

In our experiments, we analyzed how well our framework works with different AI-related writing tasks that have the potential for misuse. We focused on three main writing tasks: news writing, academic writing, and creative writing.

For the news writing task, we combined two datasets to gather human-written news articles from reputable sources such as CNN, The Washington Post, and BBC. We obtained CNN and The Washington Post articles from the Turing-Bench dataset (Uchendu et al., 2021) and BBC articles from the Xsum dataset (Narayan et al., 2018). To represent academic essays, we used the SQuAD dataset (Rajpurkar et al., 2016) and extracted Wikipedia paragraphs from the context field. For creative writing, we used the Reddit writing prompts dataset (Fan et al., 2018). After collecting the human-written text for the above tasks, we used the selected AI generators to generate corresponding AI text. Prompts used for these AI text generations can be found in Appendix A.1.1.

## 4.3 AI-generated Text Detection Setup

We followed a similar task setup to related works, considering AI-generated-text detection as a binary classification task with the labels "Human" (0) and "AI" (1). For each writing task mentioned above, we had "Human" text and "AI" text, which we split into a train, test, and validation ($\approx$ 8:1:1 ratio).

In our work, we experimented with two state-of-the-art AI text detectors representing the two main categories: zero-shot and supervised. For the zero-shot detector, we used DetectGPT (Mitchell et al., 2023), and for the supervised detector, we used a fine-tuned version of OpenAI's GPT-2 detector (OpenAI-FT) (Solaiman et al., 2019). Since the fine-tuned detector was only trained on GPT2 text, we further fine-tuned it on the text generated by each respective PLM considered in our study. More details can be found in Appendix A.1.3.

## 4.4 Baselines

We compare our method against the following recent paraphrasing-based AI-generated-text detection evasion techniques.

**Parrot paraphrasing (parrot_pp)**: PLM-based paraphrasing approach that incorporates the T5

model to paraphrase the given input text while degrading the performance of AI-generated text detection (Sadasivan et al., 2023).

**DIPPER paraphrasing (DIPPER_pp)**: PLM that is specifically catered for the task of paraphrasing. This method augments the existing paraphrasing capabilities by enabling paragraph-level paraphrasing and enabling control codes to control the diversity of the paraphrase (Krishna et al., 2023).

### 4.5 Implementation Details of EScaPe

To implement evasive soft prompt learning, we first defined the soft prompt model using the PEFT library on Huggingface[2]. We set the task type to *CAUSAL_LM*, select an initial prompt text to represent the writing task (e.g., for the news writing task, "write a news article on the given headline"), and specified the number of virtual tokens as $k = 8$. For tuning the evasive soft prompt through reinforcement learning (PPO), we utilized the TRL library (von Werra et al., 2020) for the implementation, and the framework is trained using a learning rate of $1.41 \times e^{-5}$ until the detector performance on the validation set no longer decreased. To facilitate the transfer of evasive prompts, we utilized the parameters outlined by (Su et al., 2022) in their work on transferability. Further information regarding the implementation can be found in Appendix A.1.4.

## 5 Results and Discussion

### 5.1 Evading AI Text Detectors

We recorded the results of our AI-generated-text detector evasion experiment in Table 1. The "Original" row represents the F1 score of the detector when applied to AI-text generated by the respective PLM without any modifications, while the preceding rows present the F1 score of text that has undergone different evasion methods.

**EScaPe successfully evaded detectors**

Our findings demonstrate that **EScaPe** effectively reduces the performance of the detectors across all PLMs and various writing styles. Notably, the OpenAI-FT detector experienced an average F1 score decrease of approximately 42%, while the DetectGPT detector encountered a decrease of around 22%. The discrepancy in evasion success between the two detectors may stem from their initial performance levels. For instance, DetectGPT achieved an 80% detection F1 score before evasion

[2]https://github.com/huggingface/peft

on LLaMA, Falcon, and GPT-NeoX text. Consequently, the soft prompt learned through the reward from DetectGPT is more limited compared to the soft prompt acquired through the reward from the high-performing OpenAI-FT detector. This claim can be further substantiated by analyzing the detection results of the OPT model. DetectGPT exhibits higher performance in detecting OPT-generated text, and accordingly, we observe that the soft prompt learned using the reward of DetectGPT on OPT is successful in evading the detector, unlike the cases observed with other PLMs.

**Lower False Negatives with Paraphrase Evasion**

Upon analyzing the F1 scores after evasion, it becomes apparent that **EScaPe** significantly outperforms Parrot paraphrasing. While the DIPPER paraphraser demonstrates superior evasion compared to Parrot, it falls short compared to **EScaPe** with the OpenAI-FT detector. Both paraphrasing methods show improved evasion capabilities with DetectGPT compared to OpenAI-FT. We attribute this distinction to the initial performance gap of the DetectGPT. When the detector's performance is weak for the original AI-text, specific perturbations can significantly impact its performance.

In contrast to recent works (Sadasivan et al., 2023; Krishna et al., 2023), paraphrasing exhibits limited abilities to modify the detector's performance. To address this apparent discrepancy, we analyze the underlying factors. In the presented results of Table 1, we specifically focus on the F1 score of the "AI" class, which assesses the effectiveness of the respective evasion technique in making the AI-generated text appear "Human" to the detector (false negatives). This evaluation approach differs from prior research on paraphrasing-based evasion, where both false positives and false negatives of the detector are considered, resulting in a more significant decline in the evaluation scores (AUROC) when paraphrasing is employed. However, we argue that in real-world scenarios, the most likely situation would involve malicious actors attempting to generate AI text that can evade detectors by producing false negatives.

### 5.2 Transferbility of the Evasion Prompts

We investigate two aspects of transferability: 1) the ability of **EScaPe** learned on one PLM to transfer to another, and 2) the ability of **EScaPe** learned through one detector to transfer to another detector.

| Detector | Writing Task → | News Writing | | | | Essay Writing | | | | Creative Writing | | | |
| | PLM → | LLaMA | Falcon | GPT-NEOX | OPT | LLaMA | Falcon | GPT-NEOX | OPT | LLaMA | Falcon | GPT-NEOX | OPT |
| | Method ↓ | | | | | | | | | | | | |
| OpenAI-FT | Original | 0.961 | 0.943 | 0.973 | 0.993 | 0.933 | 0.937 | 0.902 | 0.968 | 0.952 | 0.932 | 0.965 | 0.985 |
| | Parrot_PP | 0.924 | 0.903 | 0.931 | 0.972 | 0.915 | 0.918 | 0.884 | 0.933 | 0.931 | 0.911 | 0.947 | 0.962 |
| | DIPPER_PP | 0.856 | 0.811 | 0.864 | 0.824 | 0.849 | 0.802 | 0.841 | 0.811 | 0.862 | 0.806 | 0.855 | 0.835 |
| | **EScaPe** | **0.543** | **0.551** | **0.532** | **0.522** | **0.551** | **0.547** | **0.543** | **0.528** | **0.545** | **0.532** | **0.539** | **0.519** |
| DetectGPT | Original | 0.817 | 0.754 | 0.798 | 0.923 | 0.825 | 0.781 | 0.771 | 0.918 | 0.813 | 0.732 | 0.786 | 0.929 |
| | Parrot_PP | 0.732 | 0.711 | 0.705 | 0.863 | 0.757 | 0.703 | 0.691 | 0.848 | 0.788 | 0.674 | 0.718 | 0.852 |
| | DIPPER_PP | 0.635 | 0.651 | 0.658 | 0.702 | 0.641 | 0.673 | 0.647 | 0.694 | 0.628 | 0.648 | 0.663 | 0.711 |
| | **EScaPe** | **0.582** | **0.637** | **0.614** | **0.549** | **0.579** | **0.622** | **0.623** | **0.551** | **0.583** | **0.641** | **0.612** | **0.547** |

Table 1: F1 scores of the detector for text generated using various evasion techniques. 'Original' denotes text generated by the corresponding PLM without employing any evasion technique. The lowest F1 scores, indicating the highest evasion success, are highlighted in **bold**.

| Source | Target | Writing Tasks | | |
| | | News | Essay | Cre. |
| LLaMA | Falcon | 0.599 | 0.611 | 0.593 |
| | GPT-NeoX | 0.587 | 0.596 | 0.586 |
| | OPT | **0.554** | **0.559** | **0.555** |
| Falcon | LLaMA | 0.571 | 0.566 | 0.559 |
| | GPT-NeoX | 0.564 | **0.557** | **0.551** |
| | OPT | **0.561** | 0.559 | 0.554 |
| GPT-NeoX | LLaMA | 0.553 | 0.560 | 0.558 |
| | Falcon | 0.571 | 0.568 | 0.563 |
| | OPT | **0.541** | **0.552** | **0.546** |
| OPT | LLaMA | **0.572** | **0.574** | **0.571** |
| | Falcon | 0.604 | 0.613 | 0.611 |
| | GPT-NeoX | 0.588 | 0.595 | 0.583 |

Table 2: F1 scores of the detector for the text generated by the PLM in the "Target" column. Here **EScaPe** trained on the PLM in the "Source" column and transferred to the PLM in the "Target" column. The lowest F1 scores, showcasing the highest transferable success for a given "Source" PLM, are highlighted in **bold**.

### 5.2.1 Transferbility Across PLMs

Table 2 presents the transferability of **EScaPe** across different PLMs. For brevity, we provide the detection values of the OpenAI-FT detector, while the DetectGPT results can be found in Appendix A.2.1. We observe that **EScaPe** demonstrates remarkable transferable performance consistently across all writing tasks and the four PLMs investigated. For the LLaMA, Falcon, and GPT-NeoX PLMs, the deviation of transferability (maximum F1 score of transferred **EScaPe** minus the F1 score of **EScaPe** trained on itself) is less than 5%. In the case of OPT, this value is slightly higher, around 10%. Notably, **EScaPe** trained on OPT exhibits limited transferability compared to the other PLMs. One possible reason for this disparity could be the size of the language model. OPT, with 2.7B

parameters, is the smallest model examined in our study. Consequently, the **EScaPe** trained on OPT might have limitations in its capabilities to transfer to larger models. This becomes more evident when considering GPT-NeoX, the largest model we analyzed, which exhibits the lowest deviation in transferability, indicating strong transferability to all the other PLMs.

### 5.2.2 Transferbility Across Detectors

Figure 3 illustrates the extent to which **EScaPe** exhibits transferability across different detectors. For the sake of brevity, we have only presented the detection values for the news writing task, while the transferability of the detectors in the other two writing tasks can be found in Appendix A.2.1. Figure 3a reports the F1 score of the OpenAI-FT detector in two scenarios: 1) Direct - text generated using **EScaPe** trained with the reward from the OpenAI-FT detector, and 2) Transfer - text generated using **EScaPe** trained with the reward from the Detect-GPT detector. Likewise, Figure 3b depicts the F1 score of the DetectGPT detector in the direct and transfer scenarios.

Based on the two figures, it is evident that the **EScaPe** framework trained on one detector can be transferred to another detector. In both scenarios considered for the detectors, we observe that the **EScaPe** effectively evades detection, resulting in lower F1 scores ranging from 50% to 70%. Notably, we observe that the **EScaPe** trained on the supervised detector OpenAI-FT exhibits strong transferability to the zero-shot detector DetectGPT. Surprisingly, the **EScaPe** trained on OpenAI-FT yields a lower F1 score with the DetectGPT detector compared to the **EScaPe** trained on DetectGPT itself. We attribute this significant transferability to the supervised training of OpenAI-FT in detecting AI-generated text. When comparing the original per-

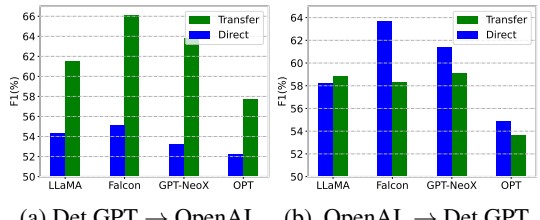

(a) Det.GPT → OpenAI.  (b) OpenAI. → Det.GPT

Figure 3: Transferability of **EScaPe** across different detectors. (a) DetectGPT → OpenAI-FT and (b) OpenAI-FT → DetectGPT. Detector $X \rightarrow Y$ denotes the evaluation of **EScaPe** trained through the reward of $X$ using the detector $Y$ ("Transfer" label). The "Direct" label denotes F1 corresponding to the detector $Y$'s performance on **EScaPe** trained through the reward of $Y$ itself.

formance (see Table 1) of OpenAI-FT and Detect-GPT, OpenAI-FT clearly outperforms DetectGPT as a detector for each respective PLM-generated text. This performance disparity is intuitive, considering that we fine-tune OpenAI-FT using the text generated by each PLM examined in our study. Consequently, **EScaPe** trained using rewards from a robust supervised detector demonstrates higher transferability.

### 5.3 Further Analysis

Here, we conduct a detailed analysis of two aspects: the quality of the generated evasive text and a comparison of different parameter-efficient tuning methods for evasive text generation.

#### 5.3.1 Quality of Evasive Text

We evaluate the perplexity change to assess the disparity between the original AI-text and the text produced after applying evasive techniques. Table 3 presents the perplexity change values for each evasion technique in LLaMA generations. The perplexity is computed using an independent PLM, GPT2-XL (Radford et al., 2019). We observe that the evasive text generated by **EScaPe** exhibits the lowest perplexity change when compared to the paraphrasing techniques, Parrot and DIPPER. This finding aligns with our expectations since we impose a KL loss between the frozen PLM and the evasive soft prompt during training. This constraint ensures that the generation, conditioned by the evasive soft prompt, remains close to the original PLM and avoids significant divergence.

#### 5.3.2 Parameter Efficient Tuning Methods

Here, we investigate the effectiveness of various parameter-efficient tuning methods, similar to the

| Method | Writing Tasks | | |
|---|---|---|---|
| | **News** | **Essay** | **Cre.** |
| Parrot_PP | 13.4 | 11.5 | 12.7 |
| DIPPER_PP | 8.1 | 7.5 | 8.3 |
| **EScaPe** | **2.5** | **1.7** | **2.1** |

Table 3: Perplexity change of the AI text after applying the evasion method. The lowest Perplexity change, indicating the highest similarity with the original AI text, are highlighted in **bold**.

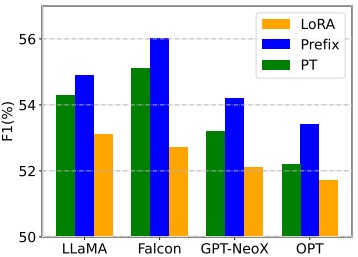

Figure 4: Evasion performance (F1 score of the detector) of **EScaPe** with different Tuning Methods

PT method, within our framework. Specifically, we explore Prefix-tuning (Li and Liang, 2021) and LoRA (Hu et al., 2021) as alternatives to PT in tuning PLM to generate evasive text. In Prefix-tuning, a set of prefix activations is prepended to each layer in the encoder stack, including the input layer. On the other hand, the LoRA method involves learning a pair of rank decompositions for the attention layer matrices while keeping the original weights of the PLM frozen. Figure 4 illustrates that LoRA yielded slightly better evasion performance compared to PT. However, it is important to note that the transferability of LoRA's rank decompositions between different PLMs has not been thoroughly studied, limiting its applicability in our objective of creating a universal evasive soft prompt.

### 6 Conclusion

In this paper, we investigated the reliability of current state-of-the-art AI-generated-text detectors by introducing a novel universal evasive soft prompt framework. Our framework, referred to as **EScaPe**, was designed to learn an evasive soft prompt capable of efficiently guiding any PLM in generating text that can effectively deceive the detectors. Through experiments conducted on various prominent PLMs across different writing tasks, our results reveal the unreliability of existing AI-generated-text detectors when confronted with text generated by PLMs guided by evasive soft prompts.

In future research, exploring the potential benefits of adversarial training in developing more robust detectors to combat the proposed evasive soft prompts would be an intriguing avenue to pursue.

# 7 Acknowledgements

This material is based upon work supported by the Office of Naval Research (ONR) under contract/grant number N00014-21-1-4002, Army Research Office under the grant number W911NF2110030, Army Research Lab under the grant number W911NF2020124 and Defense Advanced Research Projects Agency (DARPA) under the grant number HR001120C0123. The views, opinions, and/or findings expressed are those of the author and should not be interpreted as representing the official views or policies of the Department of Defense or the U.S. Government.

# 8 Limitations

## 8.1 Soft-prompting Should be Enabled

In our study, we construct the evasive soft prompt based on the assumption that the PLM is capable of using soft prompts. In other words, we consider the PLM as an open-source model, allowing us to seamlessly incorporate these learnable soft prompt embeddings into the model. This assumption restricts our framework to evaluating the AI-generated-text detectors on the PLMs that are accessible via an API, supporting only discrete natural language prompts (e.g., OpenAI API). However, if soft prompting capabilities are eventually supported through APIs in the future, our method can be applied to PLMs accessible via APIs as well.

## 8.2 Detector Reliability from the Perspective of a Malicious Actor

Generally, the reliability of AI-generated text detectors can be evaluated based on two primary criteria: 1) the misclassification of human-authored text as AI-generated, and 2) the misclassification of AI-generated text as human-authored. However, this paper focuses on the latter reliability issue, specifically addressing the detector's robustness against manipulation by malicious actors. Individuals intent on leveraging AI for plagiarism or the dissemination of misinformation will invariably explore and exploit vulnerabilities in these detection systems. We aim to address this looming threat by adopting an ethical hacking perspective to uncover potential weaknesses.

## 8.3 Evaluating the Quality of Learned Evasive Soft-prompts

Soft prompts are primarily numerical vectors, which differ significantly from hard prompt templates in interpretability aspect. This makes direct qualitative experiments on them impractical, shifting the focus to analyzing performance on the downstream tasks for which these prompts are tailored (Lester et al., 2021a; Wang et al., 2022).

# 9 Ethics Statement

## 9.1 Malicious Use of Evasive Soft Prompts

We acknowledge the potential risks associated with adversaries misusing the proposed framework to evade AI-generated text detection systems. However, we argue that the benefits of identifying limitations and vulnerabilities in state-of-the-art detector systems (red-teaming) outweigh the potential for misuse, primarily when we actively assist future researchers in addressing these issues. As a precautionary measure, we will not make the complete codebase or soft prompt weights publicly available. However, upon review, individuals or organizations engaged in legitimate research will be granted access to our framework.

## 9.2 AI-generated Text

In our work, we experiment with multiple PLMs and generate text related to news articles, academic essays, and creative writing tasks. We recognize the importance of not publicly releasing any AI-generated text used in our work, as we cannot guarantee the factual accuracy of the content. Therefore, we will implement an on-demand release structure to release our AI-generated data. Individuals or organizations requesting access to the generated data for legitimate academic research purposes will be granted permission to download the data.

## 9.3 Intended Use

It is crucial to consider the intended real-world application of **ESCaPe** and its societal impact. Our research on evasive soft prompts focuses on enabling an assessment framework for existing AI-generated text detectors. As AI-generated text becomes increasingly prevalent in various domains, the potential applications of AI-generated text detectors expand, including their role as a primary forensic tool in combating AI-generated misinformation. Therefore, assessing the detectors before their deployment becomes essential, and we emphasize that

our framework should be used for this intended purpose. However, a significant ethical concern arises if our framework is utilized for purposes other than its intended use, guiding PLMs to generate harmful and derogatory text that could negatively impact the reputation of the organizations responsible for developing the aforementioned PLMs. Hence, we strongly advise users to adhere to the intended use of our framework — only as an assessment tool for AI-generated-text detectors.

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

# A Appendix

## A.1 Reproducibility

### A.1.1 AI Generators and Prompting

Our experiments involved multiple open-source PLMs readily available on HuggingFace. Specifically, we selected the current state-of-the-art PLMs, such as LLaMA state-of-the-art PLMs such as **LLaMA** (Touvron et al., 2023)), **Falcon** (Almazrouei et al., 2023), as well as-established powerful PLMs like **GPT-NeoX** (Black et al., 2022) and **OPT** (Zhang et al., 2022). We conducted experiments using the following model sizes: LLaMA - 7 billion parameters, Falcon - 7 billion parameters, GPT-NeoX - 20 billion parameters, and OPT - 2.7 billion parameters. These PLMs were employed in two use cases. First, they were used to generate AI text for the training and testing datasets. Second, they served as the PLM in which we implemented evasive soft prompts. In both cases, the goal of the PLMs was to generate text responses given an input prompt. We set the following generation parameters: a *top-p* value of 0.96, a *temperature* of 0.9, and a *max-len* of 256 tokens. The choice of prompt used for text generation depended on the writing task. For news writing, we used the headline as the initial prompt, while for essay writing and creative writing, we selected the first 25 tokens as the prompt.

### A.1.2 Dataset Sizes

For each respective writing task, we first extracted the "human-written" portions from the sources mentioned in Section 4. We obtained 2000 samples for each writing task and divided them into train, test, and validation sets. Specifically, we allocated 100 samples for testing, 100 samples for validation, and the remaining 1800 samples for training. Subsequently, we utilized the aforementioned PLMs to generate AI-written counterparts based on the

obtained train and test data. Consequently, for each writing task, we possessed a training dataset comprising 3600 samples, a testing dataset comprising 200 samples, and a validation dataset comprising 200 samples.

### A.1.3 Detector Implementations

In our analysis, we utilized two prominent open-source detectors for AI-generated text: DetectGPT (Mitchell et al., 2023) and OpenAI's GPT-2 detector (RoBERTa-base) (Solaiman et al., 2019). **DetectGPT**: This approach also employs a proxy Pre-trained Language Model (PLM) to calculate log probabilities for individual tokens. However, its decision-making process involves comparing the log probability of the original input text with the log probability of a set of perturbed versions of the input text. These perturbations are generated using T5-base. The authors hypothesize that if the difference in log probabilities between the original text and the perturbed text consistently yields a positive value, it is likely that an AI model generated the input text. In our work, we used the original off-the-shelf implementation of DetectGPT [3].
**OpenAI-GPT2 detector**: This detector is a RoBERTa model fine-tuned on the GPT-2-output dataset [4], which consists of 250K documents from the WebText dataset (Radford et al., 2019) and 500K GPT-2-generated data. Since the OpenAI detector is trained solely on GPT-2 data, in order to use this detector with other PLMs we have, we fine-tuned a version of the OpenAI detector by combining all the training datasets from the three writing tasks. We performed fine-tuning of these detectors on an NVIDIA GeForce RTX 3090 GPU with 24 GB VRAM, setting the learning rate to $2 \times e^{-}5$ and weight decay to $0.01$.

### A.1.4 Implementation Details EScaPe

In Section 4.5, we discussed the implementation details of evasive soft prompt learning. For the implementation details of evasive soft prompt transfer, we followed the guidelines provided by (Su et al., 2022) in their cross-model soft prompt transferability work. The additional component which we didn't discuss in our main implementation details is the feedforward network we utilized as the projection network. Recall, we parameterized the projector function $g_\theta$ using a two-layer feed-forward neural network (equation 7)

---

[3]https://github.com/eric-mitchell/detect-gpt
[4]https://github.com/openai/gpt-2-output-dataset

| Source | Target | Writing Tasks | | |
|---|---|---|---|---|
| | | News | Essay | Crea. |
| LLaMA | Falcon | 0.651 | 0.643 | 0.647 |
| | GPT-NeoX | 0.659 | 0.622 | 0.636 |
| | OPT | **0.617** | **0.595** | **0.591** |
| Falcon | LLaMA | 0.697 | 0.690 | 0.684 |
| | GPT-NeoX | 0.662 | 0.665 | 0.658 |
| | OPT | **0.657** | **0.651** | **0.649** |
| GPT-NeoX | LLaMA | 0.659 | 0.644 | 0.652 |
| | Falcon | 0.671 | 0.667 | 0.678 |
| | OPT | **0.627** | **0.631** | **0.618** |
| OPT | LLaMA | **0.597** | **0.601** | **0.588** |
| | Falcon | 0.635 | 0.628 | 0.619 |
| | GPT-NeoX | 0.622 | 0.614 | 0.608 |

Table 4: F1 scores of the detector for the text generated by the PLM in the "Target" column. Here **EScaPe** trained on PLM in the 'Source' column is transferred to the "Target" column PLM. The lowest F1 scores, showcasing the highest transferable success for a given 'Source' PLM, are highlighted in **bold**.

We set the hidden size of the projector network to 768, following the same parameters defined by (Su et al., 2022), and employed the *LeakyReLU* activation function, denoted as $\sigma$.

## A.2 Additional Results

### A.2.1 Tranferbility of EScaPe

In the Experiment section, we investigated two aspects of transferability: 1) the transferability of **EScaPe** learned on one PLM to another PLM, and 2) the transferability of **EScaPe** learned on one detector to another detector. However, due to space limitations, we did not include the transferability results across PLMs for the DetectGPT detector. Additionally, we solely presented the detector transferability results for the news writing task. Table 4 presents the transferability of **EScaPe** across PLMs when trained using the reward of DetectGPT, while Tables 5 and 6 display the transferability results across detectors. These tables exhibit similar observations to those in our main Experiment section.

### A.2.2 Quality of Evasive Text

We assessed the disparity between the original AI-generated text and the text generated after applying evasive techniques in the main experiments by evaluating perplexity change. However, due to space limitations, we only presented the perplexity change for LLaMA generations. Table 7 in this section presents the complete perplexity change values for all the PLMs investigated. Perplexity is

| PLM | Writing Tasks | | |
|---|---|---|---|
| | News | Essay | Crea. |
| LLaMA | 0.615 | 0.621 | 0.619 |
| Falcon | 0.661 | 0.653 | 0.657 |
| GPT-NeoX | 0.638 | 0.644 | 0.637 |
| OPT | 0.577 | 0.581 | 0.578 |

Table 5: Transferability of **EScaPe** across different detectors. Results for the DetectGPT → OpenAI-FT transferability. **EScaPe** trained through the reward of DetectGPT, evaluated using the detector OpenAI-FT

| PLM | Writing Tasks | | |
|---|---|---|---|
| | News | Essay | Crea. |
| LLaMA | 0.589 | 0.580 | 0.588 |
| Falcon | 0.583 | 0.581 | 0.577 |
| GPT-NeoX | 0.591 | 0.599 | 0.592 |
| OPT | 0.537 | 0.540 | 0.539 |

Table 6: Transferability of **EScaPe** across different detectors. Results for the OpenAI-FT → DetectGPT transferability.**EScaPe** trained through the reward of OpenAI-FT, evaluated using the detector DetectGPT

computed using an independent PLM, GPT2-XL. Our observations align with the findings discussed earlier. The evasive text generated by **EScaPe** exhibits the lowest perplexity change compared to the paraphrasing techniques, Parrot and DIPPER. This alignment with our expectations is due to the KL loss constraint imposed during training between the frozen PLM and the evasive soft prompt. This constraint ensures that the generated text, conditioned by the evasive soft prompt, remains close to the original PLM and avoids significant divergence.

| PLM | Method | Writing Tasks | | |
|---|---|---|---|---|
| | | News | Essay | Crea. |
| LLaMA | Parrot_PP | 13.4 | 11.5 | 12.7 |
| | DIPPER_PP | 8.1 | 7.5 | 8.3 |
| | **EScaPe** | 2.5 | 1.7 | 2.1 |
| Falcon | Parrot_PP | 15.2 | 14.4 | 13.9 |
| | DIPPER_PP | 10.3 | 9.7 | 10.2 |
| | **EScaPe** | 3.3 | 2.6 | 2.8 |
| GPT-NeoX | Parrot_PP | 12.7 | 12.1 | 11.8 |
| | DIPPER_PP | 8.5 | 8.1 | 8.9 |
| | **EScaPe** | 2.3 | 1.4 | 1.9 |
| OPT | Parrot_PP | 17.3 | 16.9 | 17.7 |
| | DIPPER_PP | 11.2 | 10.4 | 9.9 |
| | **EScaPe** | 2.1 | 1.3 | 1.8 |

Table 7: Perplexity change of the AI text after applying the evasion method.