# OpenReview forum: "How Reliable Are AI-Generated-Text Detectors? An Assessment Framework Using Evasive Soft Prompts"
_EMNLP/2023/Conference — EMNLP 2023 Findings_

### Official Review · Reviewer_wSah · 2023-07-24

**Soundness:** 3

**Excitement:**

3: Ambivalent: It has merits (e.g., it reports state-of-the-art results, the idea is nice), but there are key weaknesses (e.g., it describes incremental work), and it can significantly benefit from another round of revision. However, I won't object to accepting it if my co-reviewers champion it.

**Paper Topic And Main Contributions:**

Recently, a lot of AI-generated text applications have emerged due to the powers of pre-trained language models. However, these AI-generated texts also bring a risk of misuse, and many high-performing detectors have been developed. In this paper, the authors investigate how reliable these detectors are. To answer this problem, this paper introduces a novel approach that prompts any PLM to generate texts by evading these high-performing detectors.  It utilizes a universal evasive prompt to guide PLMs in producing "human-like" text which can mislead the detectors. Extensive experiments validate the efficacy of the evasive soft prompts in their evasion of detectors.

**Reasons To Accept:**

1. This topic sounds interesting. To avoid misuse of LLMs, some applications like detectors are proposed to address detect the text whether belongs to AI generation. And this paper investigates how to evaluate the reliability of these detectors, and this motivation is great.
2. The proposed method applies reinforcement learning to optimize trainable prompt parameters and thus to mislead the detector. To extend its transferability, the authors employ projector layers to map the trainable prompt to other PLMs.
2. The writing of this paper is great.

**Reasons To Reject:**

1. Can authors explain why utilize reinforcement learning to update the prompt parameters? Why do not optimize evasive soft prompts as supervised learning (e.g., directly consider the label of the detector as the human written)?
2. The motivation of this paper is interesting. But I also want to highlight that the more important point is how evasive soft prompts can help current detectors. Therefore, it will be better for this work, if the learned soft prompt can guide the detector to detect which texts are generated by AI or not.

**Reproducibility:**

4: Could mostly reproduce the results, but there may be some variation because of sample variance or minor variations in their interpretation of the protocol or method.

**Reviewer Confidence:**

4: Quite sure. I tried to check the important points carefully. It's unlikely, though conceivable, that I missed something that should affect my ratings.

---

> ### Author Rebuttal · Authors · 2023-08-29
>
> We thank the reviewer for their thoughtful comments and constructive feedback. Please see our responses below.
>
> >**Q1)** Can authors explain why utilize reinforcement learning to update the prompt parameters? Why do not optimize evasive soft prompts as supervised learning (e.g., directly consider the label of the Detector as the human written)?
>
> **A1)** Our framework utilizes the detector's feedback (i.e., probability scores assigned to the labels 'human-written' and 'AI-generated') to guide the generation. Note that the generated text is a sequence of discrete tokens that do not allow the gradients (used for updating the text) to pass through the LLM. To bypass this differentiation problem, we propose to model the LLM equipped with our proposed soft prompts as a stochastic policy and leverage the *reinforcement learning* to directly perform the updates without the need for gradients.
>
>
>
> >**Q2)** But I also want to highlight that the more important point is how evasive soft prompts can help current detectors. Therefore, it will be better for this work, if the learned soft prompt can guide the Detector to detect which texts are generated by AI or not.
>
> **A2)** In our paper, our primary focus is to expose the vulnerability of detectors. This study is the key step to understanding the limitations of the detectors and creating defense mechanisms to robustify them against such malicious behaviors (some references). In other words, though our focus is on exposing the susceptibility of the detectors through proposing the evasive soft prompt, our study is indirectly helping the detectors to become more robust.

---

### Official Review · Reviewer_FphE · 2023-08-05

**Soundness:** 4

**Ethical Concerns:**

Yes

**Excitement:**

4: Strong: This paper deepens the understanding of some phenomenon or lowers the barriers to an existing research direction.

**Justification For Ethical Concerns:**

- Possible risks of this approach is that humans can use these evasive prompts to generate more human like texts which can evade these text detectors to generate more human like text and may possibly be misused?

**Paper Topic And Main Contributions:**

- This work proposes an adversarial technique to analyze the robustness of AI v/s Human generated text detection models.
- Their technique generates "evasive soft prompts" that can guide language models to produce adversarial text that evades detection of human v/s AI generated text. These evasive prompts are learnt through reinforcement learning.
- Once these prompts are learnt on the source model, they can be transferred to other models with further RL fine-tuning, enabling testing against future language models.
- They show that existing (AI v/s human generated text) detectors are unreliable when subjected to adversarial attacks.

**Questions For The Authors:**

A. Was there any human evaluation conducted for the text generated by these evasive prompts?
B. Was there some qualitative analysis conducted to bucketize the ways in which the evasive prompts generate more human like texts?

**Reasons To Accept:**

- The authors propose a novel approach using reinforcement learning to generate evasive soft prompts to evaluate the reliability of AI text detectors. This is an important application in these times with the increased use of generative AI.
- The authors conduct an extensive experimentation across 2 different AI text detectors, with different PLMs, baselines and datasets.
- The transferability of the evasive prompts is one strength as it enables checking future language models as well.
- The evaluation datasets cover a very relevant threat scenario.
- Experimentation is extensive and description of setup details is also precise and clear.

**Reasons To Reject:**

- No human evaluation of the text generated by these evasive soft prompts is conducted. One cannot say if the generated text actually looks like human generated text or is indistinguishable to humans.
- No examples to illustrate how this approach prompts PLMs to generate text that is more human-like


**Reproducibility:**

4: Could mostly reproduce the results, but there may be some variation because of sample variance or minor variations in their interpretation of the protocol or method.

**Reviewer Confidence:**

3: Pretty sure, but there's a chance I missed something. Although I have a good feel for this area in general, I did not carefully check the paper's details, e.g., the math, experimental design, or novelty.

---

> ### Author Rebuttal · Authors · 2023-08-29
>
> We thank the reviewer for their thoughtful comments and constructive feedback. Please see our responses below.
>
> > **Q1)** No human evaluation of the text generated by these evasive soft prompts is conducted. One cannot say if the generated text actually looks like human generated text or is indistinguishable to humans.
>
> >Was there any human evaluation conducted for the text generated by these evasive prompts?
>
> >Was there some qualitative analysis conducted to bucketize the ways in which the evasive prompts generate more human like texts?
>
> **A1)** The primary objective of the evasive soft prompting framework is to assess the reliability of current state-of-the-art detectors. We aim to guide the LLMs to produce text that successfully evades these detectors. While we cannot ensure that the generated text is indistinguishable to the human eye, our experiments confirm that it is indeed indistinguishable to the detectors. Even though we did not conduct any human evaluation or qualitative analysis of the text generated by the evasive soft prompted LLMs, as we discussed in section 5.3.1 of our paper, we employed the perplexity change measure to evaluate the quality of the text generated by our method, as utilized in related studies [1, 2]. We can gauge the magnitude of change in the generation by measuring an LLM's text perplexity before and after applying the evasive soft prompting technique. A significant difference in perplexity would indicate a considerable alteration. In our experiments, we observed that evasive soft prompting led to minimal changes in perplexity, especially when compared to paraphrasing-based perturbations introduced in [1] and [2]. This suggests that the text generated by the evasive soft-prompted LLM maintains a quality consistent with that produced by the original LLM.
>
> [1] Sadasivanet al., Can ai-generated text be reliably detected?. arXiv preprint arXiv:2303.11156.
>
> [2] Krishnaet al., Paraphrasing evades detectors of ai-generated text, but retrieval is an effective defense. arXiv preprint arXiv:2303.13408.
>
>
>
> > **Q2)**  No examples to illustrate how this approach prompts PLMs to generate text that is more human-like
>
> **A2)** Please find below some illustrative examples of evasive soft-prompted LLM generations. These examples stem from the news writing task. For these illustrations, we employed the following two hard prompts to query the LLM.
>
> *   Alton Sterling’s death appears to be another police shooting that was both legal and preventable
> *   More than 40 killed in Yemen suicide attacks
>
> **Warning:** It's important to note that these generated examples may contain misinformation. For ethical reasons, we have refrained from publicly sharing these samples.
>
> | **PLM** | **Original** | **EScaPe** |
> |:---:|:---:|:---:|
> | **Llama** | “The president’s response to this incident is entirely appropriate,” he added. “He has correctly acknowledged that race relations are a serious problem in the United States. And as he indicated, he is committed to finding a way to improve the lives of all Americans.” President Barack Obama spoke with Dallas police chief David Brown on Wednesday to express his condolences for the police officers who were gunned down in the city. He also offered his “thoughts and prayers to the families of Alton Sterling and Philando Castile, as well as to all the victims of violence across our country.” “This kind of violence is no way to make any point,” he said. “There is no justification for violence against law enforcement. None. Those who choose to violence against police officers are criminals, and thugs, and, make no mistake, they are terrorists. They are traitors. There’s no other word for it.” The president urged Americans to “come together as one united people.” President Obama said he was “heartbroken” about the senseless and tragic deaths of the police officers... | The details are still emerging, but according to the police department’s account, Sterling was shot by police after being confronted outside a Baton Rouge convenience store. Sterling’s death is the latest in a string of black men killed by police in recent weeks, and it follows other recent police shootings of African-American men that have provoked outrage from communities across the country. In the wake of those shootings, many people have called for a national conversation about race, policing, and the criminal justice system. In the wake of Sterling’s shooting, many people have called for a national conversation about race, policing, and the criminal justice system. This is a good time for that conversation, and a good time for all of us to think about what we can do to help make our communities safer and more just. People of color are disproportionately likely to be victims of crime, and people of color are disproportionately likely to be arrested and imprisoned for crimes. This means that people of color are more likely to be harmed by police officers, and they are more likely to be harmed by the people they’re trying to protect... |
> |  | A suicide bomber killed at least 40 people at a mosque in southern Yemen on Friday, while a separate blast at a funeral killed at least 10 in Sanaa, a security official said. The official said the bombing at a mosque in Lahij province killed 40 people and wounded 30. The bomber was a woman and the target was a popular Sheikh who is a prominent figure in the province, he said. Separately, a bomb blast in the capital Sanaa killed at least 10 people and wounded 40 others, the official said. He said a suicide bomber at a funeral killed those people in the Salh district of Sanaa... | The bombers struck a mosque and a residential area in a Shiite-dominated neighbourhood in the capital, Sanaa A child is brought to a hospital after an airstrike by the Saudi-led coalition, in Sanaa, Yemen, March 31, 2018. Reuters At least 40 people have been killed in a pair of suicide attacks in Yemen, including 28 at a mosque, officials said. The bombers struck a mosque and a residential area in a Shiite-dominated neighbourhood in the capital, Sanaa, on Monday. The Shiite rebels known as Houthis, who are aligned with Iran ... |
> | **Falcon** | The 37-year-old black man was shot and killed after allegedly threatening someone with a knife, according to the Baton Rouge Police Department. The officers’ body cam footage suggests otherwise. A pair of officers was dispatched to investigate a man allegedly threatening someone with a knife at the All Star Food Store in Baton Rouge, Louisiana. After arriving, the officers “engaged” with Alton Sterling, who was standing outside the store, according to the BRPD. According to the department, “one officer fired his weapon and struck Sterling and he fell to the ground.” The department said Sterling was pronounced dead at a local hospital. But the department did not specify if Sterling’s knife was ever recovered or how he was allegedly threatening someone. Local media have reported that Sterling was shot in the back, but the BRPD’s timeline does not mention this. A report from the Advocate describes Sterling as a “disturbance.” The Advocate quotes Lt. Don Kelly as saying, “one officer was involved in a disturbance and a subject was disarmed.” The Advocate spoke to several people on the scene, who said Sterling was walking when the officers arrived... | The 37-year-old, who was black, was holding a gun at the time that he was shot dead by police. However, he was not a threat to the officers who were responding to a call of a man threatening people with a gun. The tragic incident took place on July 5, 2016, in Baton Rouge, Louisiana. The shooting occurred as Sterling was walking down the street. According to the police, he was allegedly threatening people with a gun. A security camera at the scene captured the incident. The video of the incident shows that Sterling was not threatening anyone when the police arrived on the scene. In fact, he was walking away from the police and was not facing them. A local police officer, Blane Salamoni, shot Sterling. Salamoni’s partner, Howie Lake II, also shot him with his Taser. Then, they both shot him in the chest with their guns. Even though the police have a history of shooting black people, Sterling’s death has not triggered any protests... |
> |  | A suicide bomber struck near the defence ministry in the Yemeni capital Sanaa on Wednesday, killing at least 52 people and wounding 122 in the latest in a series of attacks in the country, officials said. A string of attacks on military and civilian targets have hit the country since Al-Qaeda launched a Yemeni branch of its insurgency in 2009 and the army has struggled to contain them. The attack was the deadliest in Sanaa since October 2011 when 52 people were killed in a triple bomb attack blamed on Al-Qaeda. A security official and a medic said the bomber detonated his explosives at a checkpoint in the street leading to the defence ministry, which is also home to the presidential palace. "The bomber detonated himself outside the checkpoint," a security official told AFP, adding that he was targeting "one of the key locations of Sanaa". A medic from Sanaa's main hospital, who asked not to be named, told AFP that 52 people were killed and 122 wounded. A statement from the defence ministry said "the blast took place as a suicide attacker detonated a car packed with explosives outside a checkpoint to the defence ministry". The statement said three guards were killed and two others injured... | SANAA, Yemen (AP) — Two suicide car bombs slammed into Yemen's Defense Ministry and the country's intelligence agency Tuesday, killing at least 44 people, including top security officials, in a devastating assault that raised fears of more such attacks. The Islamic State affiliate in Yemen claimed responsibility for the assault on the Defense Ministry, which was targeted by a suicide bomber who drove his explosive-laden car into the building's front gate. The attackers then stormed the ministry, leaving a large crater and a billowing cloud of smoke, before they were killed in a counter-attack by troops. IS militants also claimed responsibility for the second attack, which targeted the Sanaa headquarters of the General Intelligence Directorate, the country's main intelligence service. The country's powerful Houthi rebels, who are locked in a battle with a Saudi-led coalition for control of the country, denied any involvement in the attacks. The strikes were the deadliest in Yemen in recent months. At least 44 people were killed in the bombings, including the Interior Ministry's spokesman, the Yemeni presidency spokesman, the head of the country's national security council, the deputy head of the defense ministry and the head of the presidential guard, according to the country's government... |
> | **OPT** | The Baton Rouge police officer who shot and killed Alton Sterling has been identified as Blane Salamoni, a three-year veteran of the force.  His name was released by the East Baton Rouge Parish District Attorney’s Office Tuesday afternoon.  Salamoni responded to a call about a man with a gun at the Triple S Food Mart in Baton Rouge at approximately 5:30 p.m. on Tuesday.  Police said Salamoni shot and killed the man after he refused commands to surrender, fought with the officer, and reached for a gun in his pants.  Salamoni said on body camera footage obtained by CNN affiliate WBRZ that he was “fearing for his life” because the man threatened to shoot him and others.  But a witness who was shopping at the store at the time said Sterling never threatened the officer. The man told CNN affiliate WAFB he saw Sterling get shot.  “I’m just shocked,” said the witness, who was not identified. “I don’t know what to think, really. I just don’t understand why the officer was using lethal force.”  The video footage released by WBRZ showed Sterling pinned to the ground by a group of officers, struggling to get up, and telling the officers that he didn’t have a gun... | An autopsy shows the 44-year-old black man had six wounds to his chest and back, one of them fatal. Police said he was carrying a gun, but it was loaded with the safety off.  One officer is on paid leave; two others have been placed on administrative duty.  The case may remind some Americans of Michael Brown and Eric Garner. But the two cases are different, and the differences can be instructive.  The Garner case was about a police officer’s use of an illegal chokehold. The Brown case was about a police officer’s use of a banned chokehold, and the officer was justified in his use of deadly force. The Garner case was a tragedy. The Brown case was an avoidable tragedy.  The Alton Sterling case is also different. The officers involved in the shooting were responding to a 911 call about a man threatening people with a gun. It is unclear if Sterling was the person holding the gun. But it is clear that the officers did not know that.  The police are required to do a “reasonable” investigation when they are called to a shooting. But “reasonable” means more than “good enough.” It means more than getting the facts straight. It means getting to the bottom of what happened.  The officers say they saw Sterling fight with two people. They said Sterling grabbed one of the people and tried to take his gun. The video shows Sterling and the two men walking away from the officers. They are not fighting. They are not threatening to harm anyone. They are not even making physical contact... |
> |  | SANAA, Yemen — Yemeni officials say more than 40 people have been killed and dozens more wounded in two suicide attacks on a Shiite mosque in the country”s south. The officials say the attacks Tuesday in the town of Bayda, about 60 miles north of the city of Aden, were carried out by a suicide bomber and his two sons. They say the suicide bomber was shot by security guards, but he detonated the explosives. The second bomber then detonated his explosives next to his father, killing both. The officials spoke on condition of anonymity because they were not authorized to brief the media. Yemen”s Shiite rebels, known as Houthis, have carried out previous attacks on Shiite mosques... | At least 41 people were killed in suicide attacks in two Yemeni cities on Saturday, officials said, in the deadliest day for civilians in the country since last September.  In the southern city of Aden, security officials said two bombers blew themselves up at a military checkpoint near the city's main university, killing at least 25 people.  In the capital Sanaa, a suicide bomber set off his explosives outside the compound of the country's top military intelligence agency, killing at least 13 people, including six members of the agency.  The attacks come amid growing regional and international concern about the deteriorating security situation in Yemen, which is also being fought by Iran-backed Houthi rebels and a Saudi-led coalition.  The fighting has left more than 7,000 people dead and pushed the impoverished country to the verge of famine.  The Houthis have stepped up their attacks in recent months and claimed responsibility for the attack on Aden's airport on November 2, which killed 25 people, including a senior United Nations official... |
> | **GPT-NEOX** | The Cleveland police department has been under intense scrutiny since the Michael Brelo verdict.  Cleveland officer Timothy Loehmann shot and killed 12 year old Tamir Rice last year while he was playing with a pellet gun. The boy was holding a toy gun and never posed a threat to the officers. The only reason the officers stopped to check on Rice was because he was in a park that was known for gang activity. The officers shot and killed Tamir within two seconds of seeing him.  Rice’s death sent a shockwave through the community and across the nation. Protest erupted in Cleveland and other cities around the country. The officers were cleared of any wrongdoing.  The Brelo verdict may have finally forced the city and department to take a close look at the police department. The city has agreed to pay $6 million to the family of Brelo.  Sterling’s death occurred while he was on the ground. Police officers were called to the scene by witnesses that saw him being beaten by a group of men. The officers arrived on the scene and witnessed Sterling being pinned to the ground by a group of men. Sterling was unresponsive and he did not appear to be breathing... | On July 5, 2016, Alton Sterling was shot and killed by two police officers in Baton Rouge, Louisiana. Sterling was shot in the chest after an altercation with two police officers. The officers were responding to a call of a man in a International brand jacket selling CDs near a convenience store.  The officers were responding to a call of a man in a International brand jacket selling CDs near a convenience store.  The shooting took place in a convenience store parking lot. Surveillance video shows Sterling being tackled by one of the officers and then being shot.  Police were called to the scene because of a man selling CDs.  Sterling was shot in the chest by one of the two officers. He was taken to the hospital, but was pronounced dead.  There is no evidence that Sterling was armed or that he posed an immediate threat to the officers.  Alton Sterling’s death appears to be another police shooting that was both legal and preventable.  A legal shooting is one that is justified by the officer’s belief that his life or someone else’s life is in danger or that there is a threat of immediate bodily harm to the officer or others. A shooting is preventable when the officer could have taken other reasonable steps to avoid the shooting... |
> |  | At least 40 people have been killed and dozens more wounded in two separate suicide attacks in Yemen, officials say.  In one attack, a bomber wearing an explosives vest targeted an army base in the southern province of Shabwa, killing and wounding a number of soldiers.  Yemen's army has stepped up operations against al-Qaeda militants in the wake of a recent deadly militant assault on the security forces in the capital, Sanaa.  Meanwhile, a suicide car bomber targeted a police patrol in the city of Zinjibar, killing four people, local officials say.  The attacks come as the US is discussing how to support the Yemeni government amid reports that it is preparing for a major offensive against al-Qaeda strongholds in the country.  The US has been carrying out an air campaign against al-Qaeda in the Arabian Peninsula (AQAP) since last year... | (AP) SANAA, Yemen - More than 40 people were killed Wednesday in aoremulti-pronged suicide attacks targeting mosques and military checkpointsin northwestern Yemen. The blasts also wounded at least 50 others.  The bombings in Maarib province, which is located near the border with Saudi Arabia, underscore the challenges President Abed Rabbo Mansour Hadi faces in bringing stability to a country torn by a yearlong uprising.  There was no immediate claim of responsibility for the attacks, but al-Qaida-linked militants and separatist insurgents have exploited the chaos in Yemen. The military is increasingly stretched as it battles the rebels, who seized control of several cities in the south and parts of the west this year.  The blasts struck during the Muslim holy month of Ramadan, in which al-Qaida and other militant groups have often launched attacks.  Maarib is one of three provinces in the country where the rebellion has broken out. The other two are Abyan and Shabwa.  The U.S. Embassy in Sanaa issued a statement condemning the attacks, saying they "represent a sad day for the Yemeni people and for the Yemeni government's efforts to build national unity and stability." It also called for an investigation.  In the southern city of Aden, a suicide bomber targeted a mosque, killing at least three people and wounding a dozen, said security officials.  The blast occurred at the entrance of a mosque during the time for evening prayers, they said, speaking on condition of anonymity because they were not authorized to talk to journalists.  In the central province of Bayda, a suicide bomber killed himself and four others at a mosque after meeting with other militants, said the officials, who also spoke on condition of anonymity because they were not authorized to talk to journalists... |

---

### Official Review · Reviewer_u6Zn · 2023-08-09

**Soundness:** 3

**Excitement:**

3: Ambivalent: It has merits (e.g., it reports state-of-the-art results, the idea is nice), but there are key weaknesses (e.g., it describes incremental work), and it can significantly benefit from another round of revision. However, I won't object to accepting it if my co-reviewers champion it.

**Paper Topic And Main Contributions:**

This paper introduces an RL-driven technique for directing LLMs in producing text that can readily deceive AI-generated text detectors. The primary objective is to assess the reliability and trustworthiness of these AI-generated text detection systems. Additionally, the devised approach can be easily extended to other LLMs demonstrating similar performance levels.

**Questions For The Authors:**

1. I'd like to see more details about the F1 analysis. How many cases are false positives and false negatives?

2. Since u use soft prompts to control the output of LLMs, similarly, the generated text can be controlled to exactly match the human-written text, then the detector is meaningless. In this way, the soft prompt method seems not suitable for this problem.

**Reasons To Accept:**

1. The proposed method effectively reveals the unreliability of current AI-generated text detectors.

2. The transferability of the evasive soft prompts enables the proposed method to be a universal prompt for current and future LLMs.

**Reasons To Reject:**

1. It seems that the paper only focuses on a scenario where the AI-generated text is mistakenly identified as human-written text by the detector. It does not address the opposite case, where human-written text might be inaccurately classified as AI-generated. Both of these two situations comprise the reliability of detectors.

2. The evaluation of evasive soft prompts is missing. Although it is proven to be effective to deceive the detectors, it is hard to convince whether current evasive prompts are best or not. For example, putting Table 1 and Table 2 together, we can find that the used prompt (learned after initializing from projection) in Table 2 can not achieve comparable performance with prompts learned from scratch. Thus, how to determine whether current prompts are good enough?

3. The applications of the proposed method are limited. It must access the input embedding of used LLMs as discussed in Limitation. In addition, it is highly sensitive to the used detector and LLMs. Since it relies on the reward signal from the detector model, it requires the detector to be reliable, which seems in contrast to the motivation of this paper.

**Reproducibility:**

4: Could mostly reproduce the results, but there may be some variation because of sample variance or minor variations in their interpretation of the protocol or method.

**Reviewer Confidence:**

3: Pretty sure, but there's a chance I missed something. Although I have a good feel for this area in general, I did not carefully check the paper's details, e.g., the math, experimental design, or novelty.

---

> ### Author Rebuttal · Authors · 2023-08-29
>
> We thank the reviewer for their thoughtful comments and constructive feedback. Please see our responses below.
>
> > **Q1)** It seems that the paper only focuses on a scenario where the AI-generated text is mistakenly identified as human-written text by the detector. It does not address the opposite case, where human-written text might be inaccurately classified as AI-generated. Both of these two situations comprise the reliability of detectors.
>
> **A1)** Recently, the malicious use of AI, particularly AI-generated text, has become increasingly evident. Notable instances include AI-facilitated plagiarism [1] and the spread of misinformation using AI-generated text [2, 3]. Therefore, the most pressing reliability concern for AI-generated-text detectors lies in their possible misclassification of AI-written text as human-written. Malicious actors aiming to exploit AI for plagiarism or misinformation dissemination will invariably seek ways to exploit any vulnerabilities present in these detectors. It is precisely this imminent threat scenario that our paper wants to address, approached through an ethical hacking perspective.
>
> [1] Cotton et al., Chatting and cheating: Ensuring academic integrity in the era of ChatGPT. Innovations in Education and Teaching International, pp.1-12.
>
> [2] Yanget al., Anatomy of an AI-powered malicious social botnet. arXiv preprint arXiv:2307.16336.
>
> [3] Zhouet al., Synthetic lies: Understanding ai-generated misinformation and evaluating algorithmic and human solutions. In Proceedings of the 2023 CHI Conference on Human Factors in Computing Systems (pp. 1-20).
>
>
>
> > **Q2)** The evaluation of evasive soft prompts is missing. Although it is proven to be effective to deceive the detectors, it is hard to convince whether current evasive prompts are best or not. For example, putting Table 1 and Table 2 together, we can find that the used prompt (learned after initializing from projection) in Table 2 can not achieve comparable performance with prompts learned from scratch. Thus, how to determine whether current prompts are good enough?
>
> **A2)** It's important to emphasize that soft prompts are primarily numerical vectors, which differ significantly from hard prompt templates in interpretability. This makes direct qualitative experiments on them impractical, shifting the focus to analyzing performance on the downstream tasks for which these prompts are tailored [1, 2]. Therefore, in order to measure the quality of evasive soft prompts, we provide an extensive empirical evaluation on various benchmark datasets and detectors, demonstrating our prompts’ remarkable performance. Also, our proposed evasive soft prompts demonstrate good performance on unseen detectors, further showing that they are indeed effective.
>
> [1] Lester et al., The power of scale for parameter-efficient prompt tuning. arXiv preprint arXiv:2104.08691.
>
> [2] Wang et al., No more fine-tuning? an experimental evaluation of prompt tuning in code intelligence. In Proceedings of the 30th ACM Joint European Software Engineering Conference and Symposium on the Foundations of Software Engineering (pp. 382-394).
>
>
>
> > **Q3)** The applications of the proposed method are limited. It must access the input embedding of used LLMs as discussed in Limitation. In addition, it is highly sensitive to the used detector and LLMs. Since it relies on the reward signal from the detector model, it requires the detector to be reliable, which seems in contrast to the motivation of this paper.
>
> **A3a)** Access to the input embedding is primarily a constraint only when dealing with closed-source proprietary LLMs. However, given the rising popularity and impact of open-source LLMs like Llama 2, we see a growing adoption of open-source LLMs for various writing tasks. Hence, we believe that having methods to evaluate the reliability of detectors against malicious threats of these open-source LLMs is both critical and essential.
>
> **A3b)** The proposed evasive soft prompts can be learned with the feedback of any known AI-generated-text detector on any open-source LLM. Through our experiments, we have demonstrated that the evasive soft prompts learned from a known detector can be effectively used to evade an unknown detector. This is shown in Figure 3 on page 8. We have also proposed a method for transferring the evasive soft prompts across LLMs and have empirically shown its effectiveness in Table 2. Thus, we disagree with the notion that evasive soft prompts are sensitive to the used detector and LLM.
>
> **A3c)** The primary aim of the evasive soft prompt is to evaluate the robustness of state-of-the-art (SOTA) AI-generated-text detectors when confronted with malicious users attempting to evade AI-text detection. These detectors are known for their accuracy and reliability in standard testing scenarios, where they consistently distinguish between AI-generated and human-written text. Our research advances this understanding by exposing potential vulnerabilities of these SOTA techniques when facing malicious usage of LLMs, approached through an ethical hacking lens. Given this context, we contest the notion that relying on generally reliable detectors could be considered a limitation in our study.
>
>
>
> > **Q4)**  I'd like to see more details about the F1 analysis. How many cases are false positives and false negatives?
>
> **A4)** As detailed in Section 5.1 (line 506) of our paper, our primary emphasis is on the F1 score for the "AI" class. This metric directly reflects our research objective: to measure the efficacy of evasive soft prompts in making AI-generated text appear "Human" to the detector. Therefore, the ground truth label is always 1 ("AI") since we are evaluating the detector exclusively on the text generated by the evasive soft-prompted LLMs. In this context, the notion of False Positives is not applicable as there are no class 0 samples that could be misclassified as class 1. Kindly refer to the table below for false negative values (calculated from approximately 200 samples from both the test and validation sets for a specified writing task).
>
> 1.  TN values for the Table 1 of the paper
>
> | Detector. | Writing Task | News Writing |        |          |     | Essay Writing |        |          |     | Creative Writing |        |          |     |
> |-----------|--------------|--------------|--------|----------|-----|---------------|--------|----------|-----|------------------|--------|----------|-----|
> |           | **PLM ->**          | LLaMA        | Falcon | GPT-NEOX | OPT | LLaMA         | Falcon | GPT-NEOX | OPT | LLaMA            | Falcon | GPT-NEOX | OPT |
> |           |              |              |        |          |     |               |        |          |     |                  |        |          |     |
> | **OpenAI-FT** | Original     | 15           | 21     | 10       | 3   | 25            | 23     | 35       | 12  | 18               | 25     | 13       | 6   |
> |           | Parrot_PP    | 28           | 34     | 25       | 11  | 31            | 30     | 41       | 25  | 26               | 33     | 20       | 15  |
> |           | DIPPER_PP    | 49           | 62     | 47       | 58  | 52            | 65     | 54       | 63  | 48               | 65     | 50       | 56  |
> |           | EScaPE       | 122          | 121    | 124      | 126 | 123           | 123    | 124      | 127 | 124              | 127    | 126      | 129 |
> | **DetectGPT** | Original     | 60           | 77     | 66       | 28  | 59            | 71     | 74       | 30  | 63               | 84     | 70       | 26  |
> |           | Parrot_PP    | 82           | 87     | 89       | 47  | 77            | 91     | 93       | 52  | 70               | 98     | 88       | 51  |
> |           | DIPPER_PP    | 104          | 101    | 99       | 90  | 105           | 98     | 103      | 93  | 108              | 104    | 100      | 89  |
> |           | EScaPE       | 115          | 104    | 109      | 121 | 117           | 109    | 108      | 123 | 117              | 105    | 111      | 124 |
>
>
> 2.  TN values for the Table 2 of the paper
>
> |            |            | Writing Task |       |       |
> |------------|------------|--------------|-------|-------|
> | **Source PLM** | **Target PLM** | **News** | **Essay** | **Crea.** |
> | **LLaMA**      | Falcon     | 112          | 111   | 115   |
> |            | GPT-NeoX   | 114          | 114   | 117   |
> |            | OPT        | 120          | 121   | 123   |
> | **GPT-NeoX**   | LLaMA      | 120          | 121   | 122   |
> |            | Falcon     | 117          | 119   | 121   |
> |            | OPT        | 123          | 123   | 124   |
> | **OPT**        | LLaMA      | 117          | 118   | 119   |
> |            | Falcon     | 111          | 110   | 111   |
> |            | GPT-NeoX   | 114          | 114   | 117   |
> | **Falcon**     | LLaMA      | 117          | 120   | 122   |
> |            | GPT-NeoX   | 118          | 122   | 123   |
> |            | OPT        | 119          | 121   | 123   |
>
>
>
> 3. TN values for the Figure 3
>
>
>
> *   DetectGPT -> OpenAI FT
>
> |          | Writing Task |       |       |
> |----------|--------------|-------|-------|
> | **PLM**      | **News**         | **Essay** | **Crea.** |
> | LLaMA    | 108          | 107   | 108   |
> | Falcon   | 99           | 100   | 100   |
> | GPT-NEOX | 104          | 102   | 104   |
> | OPT      | 116          | 115   | 116   |
>
> *   OpenAI FT -> DetectGPT
>
> |          | Writing Task |       |       |
> |----------|--------------|-------|-------|
> | PLM      | News         | Essay | Crea. |
> | LLaMA    | 114          | 115   | 114   |
> | Falcon   | 115          | 115   | 116   |
> | GPT-NEOX | 113          | 112   | 113   |
> | OPT      | 123          | 123   | 123   |
>
>
>
> > **Q5)** Since u use soft prompts to control the output of LLMs, similarly, the generated text can be controlled to exactly match the human-written text, then the detector is meaningless. In this way, the soft prompt method seems not suitable for this problem.
>
> **A5)** We find the statement "similarly, the generated text can be controlled to exactly match the human-written text" in the review somewhat ambiguous. If the reviewer is suggesting a scenario where a malicious actor orchestrates controlled text generation using a PLM to emulate the intricacies of human-written text with the intent of de, then that is precisely the behavior our proposed evasive soft prompt aims to simulate. Our intention is to highlight the vulnerabilities of existing detectors when confronted with such threat scenarios.

---

### Meta-Review · Area_Chair_6ADv · 2023-09-18

**Recommendation:** 3

**Metareview:**

This paper presents a reinforcement learning-based technique to direct language models (LLMs) in generating text that can deceive AI-generated text detectors. The aim is to evaluate the reliability and trustworthiness of these detection systems. The proposed approach is transferable to other LLMs with similar performance levels.

Strengths:
* The paper addresses an important and timely issue, assessing the reliability of AI-generated text detectors.
* The proposed method effectively reveals the unreliability of current AI-generated text detectors.
* The transferability of the evasive soft prompts makes the proposed method applicable to current and future LLMs.
* The paper is well-written and has extensive experiments across different AI text detectors, baselines, and datasets.

Weaknesses:
* The paper does not fully address the case where human-written text is incorrectly classified as AI-generated, affecting the overall reliability of detectors.
* The evaluation of evasive soft prompts is missing, making it difficult to determine the effectiveness of the current prompts.
* The paper's applications are limited, as it requires access to input embeddings of used LLMs and is sensitive to the used detector and LLMs.
* No human evaluation of the text generated by the evasive soft prompts is conducted, so the actual resemblance to human-generated text remains unclear.

Overall, this paper has merits and makes significant contributions to the field. However, it has some key weaknesses that may benefit from further revisions. The topic is interesting, and the motivation is strong, but addressing the concerns raised by the reviewers could further strengthen the paper.

---

### Decision · Program_Chairs · 2023-10-07

**Decision:**

Accept-Findings

**Comment:**

This paper presents a reinforcement learning-based technique to direct language models (LLMs) in generating text that can deceive AI-generated text detectors. The aim is to evaluate the reliability and trustworthiness of these detection systems. The proposed approach is transferable to other LLMs with similar performance levels.

Strengths:
* The paper addresses an important and timely issue, assessing the reliability of AI-generated text detectors.
* The proposed method effectively reveals the unreliability of current AI-generated text detectors.
* The transferability of the evasive soft prompts makes the proposed method applicable to current and future LLMs.
* The paper is well-written and has extensive experiments across different AI text detectors, baselines, and datasets.

Weaknesses:
* The paper does not fully address the case where human-written text is incorrectly classified as AI-generated, affecting the overall reliability of detectors.
* The evaluation of evasive soft prompts is missing, making it difficult to determine the effectiveness of the current prompts.
* The paper's applications are limited, as it requires access to input embeddings of used LLMs and is sensitive to the used detector and LLMs.
* No human evaluation of the text generated by the evasive soft prompts is conducted, so the actual resemblance to human-generated text remains unclear.

Overall, this paper has merits and makes significant contributions to the field. However, it has some key weaknesses that may benefit from further revisions. The topic is interesting, and the motivation is strong, but addressing the concerns raised by the reviewers could further strengthen the paper.